

# Relationship between the Dark Triad and depressive symptoms

Raquel Gómez-Leal[1], Alberto Megías-Robles[1],
María José Gutiérrez-Cobo[2], Rosario Cabello[2],
Enrique G. Fernández-Abascal[3] and Pablo Fernández-Berrocal[1]

[1] Department of Basic Psychology, University of Málaga, Málaga, Spain
[2] Department of Developmental and Educational Psychology, University of Granada, Granada, Spain
[3] Faculty of Psychology, Universidad Nacional de Educación a Distancia, Madrid, Spain

## ABSTRACT

The Dark Triad (DT) is composed of three closely related personality traits: psychopathy, Machiavellianism and narcissism. These traits have been linked to emotional deficits. The aim of the present study was to analyze the relationship between the DT traits, including sub-dimensions, and depressive symptoms in order to identify those factors most strongly associated with the development of depression in individuals scoring high on DT. For these purposes, a total of 791 adults ($M = 35.76$ years; 24.91% males) completed a questionnaire battery including DT traits and depression measures. A positive significant correlation was found between psychopathy and Machiavellianism traits (total score and all sub-dimensions) and depressive symptoms. For narcissism, the direction of the correlation was dependent on the sub-dimension assessed. A model explaining 26.2% of the depressive symptoms scores was composed of the callous affect and criminal tendencies sub-dimensions of psychopathy, cynical view of human nature, which is a sub-dimension of Machiavellianism, and entitlement and self-sufficiency, which are sub-dimensions of narcissism. In addition, some of the relationships found between DT sub-dimensions and depressive symptoms appeared to depend on gender. Our results could have implications for detection and intervention programs aimed at decreasing the negative emotional consequences suffered by individuals with high DT scores. Limitations and future lines of research are discussed.

## INTRODUCTION

The term Dark Triad (DT) describes a set of three distinct but related sub-clinical personality traits: psychopathy, Machiavellianism and narcissism (*Paulhus & Williams, 2002*). These traits have often been associated with negative aspects of personality; for instance, psychopathy is related to higher levels of aggressiveness and impulsivity (*Kennealy et al., 2010*), Machiavellianism is linked to hypocrisy and manipulation (*Paulhus & Williams, 2002*), and narcissism is linked to dominance, superiority and egocentric attitude (*Paulhus & Williams, 2002*). Among these undesirable outcomes we can also find emotional deficits such as anxiety or low empathy associated with the three

Corresponding author
Alberto Megías-Robles,
amegias@uma.es

DT traits (*Megías et al., 2018*; *Miller et al., 2010*; *Jonason & Kroll, 2015*), as well as difficulty in regulating mood and alexithymia associated with psychopathy and Machiavellianism (*Miao et al., 2019*; *Cairncross et al., 2013*). The aim of the current study was to explore how the three personality traits comprising the DT could be related to depressive symptoms.

Depression is one of the most common mental disorders worldwide (*Lépine & Briley, 2011*). This disorder is considered a major public health problem because of the consequences it entails, not only for those affected and their family members but also for society, with very high economic costs in terms of the use of services and loss of productivity (*Cassano & Fava, 2002*). The most common depressive symptoms are continuous depressed mood, and somatic and cognitive changes that affect the individual on a day-to-day basis such as a general loss of interest, low self-esteem, irritability, or changes in appetite and sleep (*American Psychiatric Association, 2013*). Moreover, research has revealed that women, in comparison with men, show a higher prevalence, incidence, and morbidity risk of depressive symptoms, beginning at mid-puberty and persisting through adult life (*Piccinelli & Wilkinson, 2000*).

Psychopathy, Machiavellianism and narcissism are all encompassed by the DT construct because they share a common core of important characteristics such as callous-manipulation, tendencies towards self-promotion, and the performance of norm-violating acts, deception, and aggressiveness (*Furnham, Richards & Paulhus, 2013*; *Paulhus & Williams, 2002*). However, despite the overlap between these traits, each has their own unique particularities and is composed of different sub-dimensions. Psychopathy is characterized by thrill-seeking behaviors, dishonesty, egocentricity, manipulation, and antisocial behavior (*Hare, 1993*); Machiavellianism is characterized by exploitative behavior, insincerity, and callousness (*Christie & Geis, 1970*); and narcissism is characterized by exhibitionism, feelings of superiority, dominance and admiration seeking (*Morf & Rhodewalt, 2001*; *Raskin & Terry, 1988*).

Recent years have seen an exponential rise in the level of attention paid to studying DT (*Watts et al., 2017*; *Muris et al., 2017*). Although there is a specific literature showing the advantages of scoring high on DT traits, such as greater confidence, commitment, control and physical activity (*Sabouri et al., 2016a*, *2016b*), much of the research on DT has focused on the negative consequences that these traits may have for society in general (*Kennealy et al., 2010*). However, relatively little importance has been assigned to the negative emotional consequences that may be suffered by individuals scoring high on the DT. We believe it is important to consider not only those outcomes that affect society, but also the personal consequences that may affect individuals with high DT scores, such as depressive symptoms. In this regard, a number of previous studies have explored the relationship between depression and some of the three DT personality traits separately (*Al Aïn et al., 2013*; *Stinson, Becker & Tromp, 2005*), but, in order to provide a better understanding of this relationship, it is necessary to characterize the DT term that encompasses a concurrent analysis of the three personality traits and their sub-dimensions (*Jones & Paulhus, 2017*).

Research analyzing the relationship between depressive symptoms and psychopathy have yielded mixed results. Whilst some studies have found a negative relationship between depression and psychopathy (*Stålenheim & Von Knorring, 1996*), others have found a positive relationship between these constructs (*Stinson, Becker & Tromp, 2005*; *Tokarev et al., 2017*). For instance, *Stinson, Becker & Tromp (2005)* observed that depressive symptoms are common in extreme psychopathic groups. Second, in relation to Machiavellianism, although there are also some conflicting findings, these tend to be more consistent in showing a positive relationship between both concepts. For example, *Al Aïn et al. (2013)* found a positive correlation between the total score on the Machiavellianism IV Scale (*Christie & Geis, 1970*) and the total score on the Beck Depression Inventory II (*Beck, Steer & Brown, 1996*). Finally, research focused on narcissism has shown that there appears to be no relationship between this trait and depression (*Jonason et al., 2015*). This could be due to the fact that narcissism—although associated with maladaptive behavior—is also related to subjective wellbeing (*Rose & Campbell, 2004*). To our knowledge, only one study has explored the relationship between the three DT traits and depression (*Jonason et al., 2015*). In this study, *Jonason et al. (2015)* found a positive relationship between depression and the traits of psychopathy and Machiavellianism. However, this study did not consider the different sub-dimensions of the DT traits and it is focused on general measures of wellbeing. Given that each sub-dimension is related to different characteristics, it is important to analyze each of these separately in order to draw more specific conclusions and to fully understand the possible clinical and social implications of the findings. For this reason, in the present study we focused on all the sub-dimensions of the DT traits.

In addition, previous research has revealed gender differences in the personality traits comprising the DT. Firstly, in relation to psychopathy, both in clinical and community samples, this trait has shown to be much stronger in men than in women (*Cale & Lilienfeld, 2002*; *Nicholls et al., 2005*). Secondly, although there are few studies analyzing the relationship between Machiavellianism and gender, these have also shown higher scores for men than women (*Krampen et al., 1990*). Finally, through a meta-analysis in which 355 studies were included, *Grijalva et al. (2015)* found that men scored higher than women on narcissism. All of these previous findings emphasize the importance of incorporating gender differences in the study of DT traits.

A more detailed analysis of the relationship between depressive symptoms and the sub-dimensions of the DT traits that considers gender effects could help professionals and researchers to establish better emotion-oriented treatments in people with high scores on DT traits. This issue is of importance because the comorbidity of high scores on DT traits and depression could prolong the time needed to respond to depression treatment (*Mulder, Joyce & Luty, 2003*), lead to a higher frequency of relapses (*Ramklint & Ekselius, 2003*) or even result in extreme cases such as suicide, since depression and the characteristics of non-adaptive personalities are related to suicidal ideas (*Nock et al., 2010*).

The main objective of this study was to analyze the relationship between the personality traits associated with DT and the possibility of presenting depressive symptoms.

To carry out a detailed study of this relationship, we focused on each of the sub-dimensions of each DT trait in an attempt to develop a model that helps to identify those DT factors that are most strongly associated with depressive symptoms. In addition, given the previously demonstrated gender differences in DT traits, we also examined whether the relationship between these traits and the sub-dimensions of depression are equal across genders. We hypothesize that (1) there are gender differences in the total scores and sub-dimensions of the DT traits; (2) there is a positive relationship between depressive symptoms and the psychopathy and Machiavellianism traits; (3) there is no significant relationship between depressive symptoms and narcissism; (4) the relationships between depressive symptoms and DT traits are moderated by gender; and (5) depressive symptoms could partially be related to the joint effect of a set of sub-dimensions of the DT traits.

## METHODS

### Participants

Seven hundred and ninety-one volunteer participants from the Spanish National Distance Education University took part in this study. The age of the sample ranged from 18 to 66 years, with a mean of 35.76 years (SD = 9.54). One hundred and ninety-seven of the participants were men (24.91%). All participants were informed that confidentiality and anonymity of the collected data would be assured, and all were treated in accordance with the Helsinki declaration (*World Medical Association, 2008*). The Research Ethics Committee of the University of Málaga approved the study protocol as part of the project SEJ-07325 (IRB approval number 10-2018-H).

### Procedure and instruments

Participants completed the depression and DT questionnaires online through LimeSurvey platform (http://limesurvey.org). They needed approximately 30 min to complete the whole study. Access to the questionnaires was provided via email invitation from the authors. Informed consent was found on the first page of the online questionnaire, where they were assured of the confidentiality of their responses. The questionnaires were set up so that empty responses were not allowed in order to avoid the possibility of missing data. A description of each scale is detailed below.

*Beck Depression Inventory II* (BDI-II; *Beck, Steer & Brown, 1996*). The BDI is a 21-item self-report scale widely used to assess depression. It corresponds with symptoms of depressive disorders as listed in the Diagnostic and Statistical Manual of Mental Disorders (*American Psychiatric Association, 2013*). Each item contains four self-evaluative statements ordered in increasing severity of the depressive symptoms (scores for each item range from 0 to 3). Individuals must choose the statement that best reflects their feelings over the past week. The BDI provides a continuous score of depressive symptoms ranging from 0 to 63. A score of 0–13 indicates minimal depression, 14–19 indicates mild depression, 20–28 moderate depression, and 29–63 severe depression (*Beck, Steer & Brown, 1996*). We used the Spanish version of the scale (*Sanz, Perdigón & Vázquez, 2003*). In our study, the scale showed excellent internal consistency (ordinal omega = 0.95).

*The 34-item Self-Report Psychopathy Scale-III* (SRP-III; *Mahmut et al., 2011*). The SRP-III is a 34-item self-report measure of psychopathic traits based on a four-factor structure model of psychopathy (*Hare & Neumann, 2005*; *Vitacco, Neumann & Jackson, 2005*). The questionnaire includes a total score and four sub-dimension scores: interpersonal manipulation (IPM), criminal tendencies (CT), erratic lifestyle (ELS), and callous affect (CA). Responses are given on a 5-point Likert type scale ranging from 1 ("Disagree strongly") to 5 ("Agree strongly"). The Spanish version of the questionnaire was used in our study (*Gómez-Leal et al., 2019*). Internal consistency for the sample of the present study was good (ordinal omega of the total score was 0.92, whilst for the sub-dimensions this ranged between 0.77 and 0.89).

*Machiavellianism-IV Scale* (MACH-IV; *Christie & Geis, 1970*). The MACH-IV is a 20-item self-report measure. Ten items are related to high Machiavellianism and 10 items to low Machiavellianism. Participants are requested to rate the extent to which they agreed or disagreed with the statement of each item using a 6-point Likert scale ranging from 1 ("Strongly Disagree") to 6 ("Strongly Agree"). Mach-IV contains a total score and three sub-dimensions: interpersonal tactics (T), cynical view of human nature (V) and disregard for conventional morality (M). One of the items in the M sub-dimension was removed because it is outdated, according to suggestions in the previous literature (*Rauthmann, 2013*). We used the Spanish version of the questionnaire (*Rada, de Lucas Taracena & Rodríguez, 2004*). Internal consistency for the sample of the present study was adequate (ordinal omega of the total score was 0.79; whilst for the sub-dimensions this ranged between 0.70 and 0.72).

*Narcissistic Personality Inventory* (NPI; *Raskin & Hall, 1979*; *Raskin & Terry, 1988*). The NPI consists of 40 forced-choice items, where the participants must choose between two alternative statements, scoring 1 point if they choose the narcissistic statement and 0 points if they choose the non-narcissistic statement. NPI includes a total score and seven sub-dimensions: authority (AUT), self-sufficiency (SS), vanity (VAN), exploitativeness (EXP), superiority (SUP), exhibitionism (EXH) and entitlement (ENT). We used the Spanish version of the questionnaire (*García & Cortés, 1998*). Internal consistency for the sample of the present study was adequate (ordinal omega of the total score was 0.87, whilst for the sub-dimensions this ranged between 0.67 and 0.81).

## Data analysis

In order to study the relationship between the DT and depression, we conducted the following analyses. First, descriptive statistics were carried out to explore the characteristics of the total scores and all the sub-dimensions of the measures employed. Moreover, possible gender differences for each variable were explored using Mann–Whitney $U$ tests. Second, the relationship between depression and DT was evaluated using Spearman's correlation analysis, both for the whole sample (men and women) and divided according to gender. Mann–Whitney $U$ tests and Spearman's correlations were used given that the normality assumption was rejected for the variables included in the study (Kolmogorov–Smirnov test < 0.05). Third, we explored possible gender differences in the significant correlations found in the previous step using Fisher's

Z-test. Descriptive statistics, Mann–Whitney $U$ tests, Spearman's correlations, and Fisher's Z-test were conducted using the SPSS® version 23.0 (IBM Corporation, Armonk, NY, USA) and FZT computator (http://psych.unl.edu/psycrs/statpage/regression.html). The alpha level was set at 0.05 for all the analyses.

Finally, we carried out a generalized additive model (GAM; *Hastie & Tibshirani, 1990*; *Wood, 2017*) in order to identify the sub-dimensions of the DT traits most strongly associated with depression. The GAM allows for modeling complex relationships between variables through both linear and non-linear, parametric and non-parametric smoothing functions combined in the same model. This semi-parametric approach was used, since normality and homoscedasticity assumptions were rejected. The values of the residuals were not normally distributed (Shapiro–Wilks test: $p < 0.05$) and their variance was not constant (checked by visual inspection of the scatter plot showing the standardized residuals against the standardized predicted values). The multicollinearity assumption was not violated (always VIF ≤ 1.79 and tolerance ≥ 0.65). We introduced into the model those DT sub-dimensions that were significant for the whole sample in the correlational analysis. Each of these sub-dimensions was included as a smooth term. The main effect of gender was introduced as a predictor to control for the possible influence of this variable. Total scores were not included, given that these are the result of averaging their sub-dimensions. The basis dimension used to represent the smooth terms was determined automatically using penalized regression splines by the mgcv package (*Wood, 2017*). The GAM analysis was conducted using the statistical software R 3.6.1. (*R Core Development Team, 2000*) with the mgcv package (*Wood, 2017*). The alpha level was set at 0.05.

## RESULTS

Table 1 shows the descriptive statistics and gender differences for the variables included in the study (total scores and sub-dimensions). We observed that men, in comparison with women, showed higher levels of psychopathy on the total score and all its sub-dimensions (except in the callous affect sub-dimension), higher levels of Machiavellianism on the total score and the interpersonal tactics and disregard for conventional morality sub-dimensions, and higher levels of narcissism on the vanity and entitlement sub-dimensions.

Spearman's correlation analysis conducted for DT and depressive symptoms for the whole sample (men and women) revealed that the higher the psychopathy and Machiavellianism scores (total score and all sub-dimensions), the higher the depressive symptoms. In the case of narcissism, higher scores on the self-sufficiency sub-dimension were related to fewer depressive symptoms, whilst higher scores on the entitlement sub-dimension were related to more depressive symptoms. When the sample was divided by gender, a similar pattern of results was found, with the exception that men showed a positive relationship between the exhibitionism sub-dimension of narcissism and depressive symptoms, and women showed a negative relationship between the vanity and superiority sub-dimensions of narcissism with depressive symptoms. In addition, women did not show a relationship between the disregard for conventional morality sub-dimension of Machiavellisanism and depressive symptoms (see Table 2). When

**Table 1 Means, standard deviations (SD), and Mann–Whitney _U_ test with effect size (_r_) for gender differences.**

| | Global sample | | Men | | Women | | | |
|---|---|---|---|---|---|---|---|---|
| | Mean | SD | Mean | SD | Mean | SD | Mann–Whitney _U_ | Effect size (_r_) |
| BDI Total | 8.42 | 8.57 | 7.48 | 7.80 | 8.73 | 8.80 | 54,537 | 0.05 |
| SRP-III Total | 61.04 | 14.06 | 66.18 | 16.26 | 59.34 | 12.82 | 42,988** | 0.20 |
| MACH-IV | 67.10 | 13.61 | 69.01 | 14.81 | 66.47 | 13.15 | 52,739* | 0.07 |
| NPI Total | 11.96 | 5.05 | 12.39 | 5.30 | 11.82 | 4.96 | 54,869 | 0.04 |
| IPM (SRP-III) | 14.76 | 4.68 | 16.47 | 5.18 | 14.19 | 4.35 | 41,895** | 0.21 |
| CT (SRP-III) | 15.31 | 5.55 | 17.27 | 6.56 | 14.66 | 5.01 | 43,674** | 0.19 |
| ELS (SRP-III) | 17.37 | 5.36 | 18.39 | 5.82 | 17.03 | 5.16 | 50,516** | 0.10 |
| CA (SRP-III) | 13.61 | 3.32 | 14.05 | 3.52 | 13.46 | 3.24 | 53,229 | 0.07 |
| T (MACH-IV) | 28.05 | 7.39 | 29.26 | 7.88 | 27.65 | 7.18 | 51,335** | 0.09 |
| V (MACH-IV) | 29.31 | 7.10 | 29.95 | 7.29 | 29.10 | 7.02 | 54,665 | 0.05 |
| M (MACH-IV) | 1.51 | 0.91 | 1.67 | 1.07 | 1.46 | 0.84 | 52,679* | 0.09 |
| AUT (NPI) | 3.08 | 1.47 | 3.19 | 1.53 | 3.04 | 1.45 | 54,846 | 0.05 |
| SS (NPI) | 2.54 | 1.25 | 2.58 | 1.19 | 2.53 | 1.27 | 56,593 | 0.03 |
| VAN (NPI) | 0.80 | 0.93 | 0.66 | 0.92 | 0.84 | 0.93 | 51,280** | 0.10 |
| EXP (NPI) | 1.40 | 1.24 | 1.51 | 1.26 | 1.36 | 1.24 | 54,391 | 0.05 |
| ENT (NPI) | 1.24 | 1.06 | 1.43 | 1.14 | 1.18 | 1.02 | 51,525** | 0.09 |
| SUP (NPI) | 1.61 | 1.07 | 1.75 | 1.16 | 1.56 | 1.03 | 53,441 | 0.07 |
| EXH (NPI) | 1.30 | 1.48 | 1.26 | 1.51 | 1.32 | 1.47 | 56,280 | 0.03 |

Notes:
SRP-III (IPM, interpersonal manipulation; CT, criminal tendencies; ELS, erratic lifestyle; CA, callous affect); MACH-IV (T, interpersonal tactics; V, cynical view of human nature; M, Disregard for conventional morality); NPI (AUT, authority, SS, self-sufficiency; VAN, vanity; EXP, exploitativeness; ENT, entitlement; SUP, superiority; EXH, exhibitionism).
\* $p < 0.05$.
\*\* $p < 0.01$.

**Table 2 Spearman's correlations between (sub)dimensions of DT and depression.**

| | SRP-III Total | MACH-IV Total | NPI Total | IMP (SRP-III) | CT (SRP-III) | ELS (SRP-III) | CA (SRP-III) | T (MACH-IV) | V (MACH-IV) | M (MACH-IV) | AUT (NPI) | SS (NPI) | VAN (NPI) | EXP (NPI) | SUP (NPI) | EXH (NPI) | ENT (NPI) |
|---|---|---|---|---|---|---|---|---|---|---|---|---|---|---|---|---|---|
| Whole sample | 0.27** | 0.30** | −0.03 | 0.20** | 0.20** | 0.14** | 0.29** | 0.24** | 0.30** | 0.09* | 0.02 | −0.28** | −0.07 | −0.01 | −0.05 | 0.06 | 0.14** |
| Men | 0.32** | 0.38** | 0.10 | 0.23** | 0.30** | 0.15* | 0.25** | 0.30** | 0.34** | 0.22** | 0.02 | −0.16* | 0.07 | 0.09 | 0.05 | 0.15* | 0.20** |
| Women | 0.28** | 0.29** | −0.07 | 0.22** | 0.20** | 0.14** | 0.32** | 0.23** | 0.29** | 0.05 | 0.03 | −0.32** | −0.12** | −0.03 | −0.08* | 0.04 | 0.13** |

Notes:
SRP-III (IPM, interpersonal manipulation; CT, criminal tendencies; ELS, erratic lifestyle; CA, callous affect); MACH-IV (T, interpersonal tactics; V, cynical view of human nature; M, disregard for conventional morality); NPI (AUT, authority; SS, self-sufficiency; VAN, vanity; EXP, exploitativeness; ENT, entitlement; SUP, superiority; EXH, exhibitionism).
\* $p < 0.05$.
\*\* $p < 0.01$.

specific gender comparisons were made with respect to the correlations using the Fisher's _z_-test, it was found that women, compared with men, showed a stronger negative relationship between the narcissism sub-dimensions of self-sufficiency ($Z = 2.06$, $p < 0.05$) and vanity ($Z = 2.31$, $p < 0.05$) with depressive symptoms. Moreover, men, compared

with women, showed a stronger positive relationship between the Machiavellianism sub-dimension of disregard for conventional morality ($Z = 2.10$, $p < 0.05$) and depressive symptoms.

Finally, the GAM analysis revealed five significant sub-dimensions, which explained 26.3% of the total variance. The significant sub-dimensions were criminal tendencies (estimated df = 6.95, $F = 5.31$, $p < 0.001$) and callous affect (estimated df = 1.00, $F = 10.83$, $p = 0.001$) from psychopathy, cynical view of human nature (estimated df = 1.86, $F = 7.28$, $p < 0.001$) from Machiavellianism, and self-sufficiency (estimated df = 2.75, $F = 19.83$, $p < 0.001$) and entitlement (estimated df = 1.00, $F = 4.57$, $p = 0.03$) from narcissism. Figure 1 presents the shape of the significant relationships. According to the estimated degree of freedom (1 df corresponds to a linear fit) and graphic representation, the relationships between depressive symptoms and the callous affect and entitlement sub-dimensions were linear and showed a positive association. The relationships between depressive symptoms and each of the sub-dimensions of criminal tendencies, cynical view of human nature, and self-sufficiency were all non-linear (df larger than 1). However, despite the non-linear relationship, both criminal tendencies and cynical view of human nature also showed a general tendency to increase along with the increase in depressive symptom scores, particularly for the highest scores on criminal tendencies. With respect to self-sufficiency, a negative relationship was observed with depressive symptoms, except for the highest scores on self-sufficiency.

## DISCUSSION

The main aim of this study was to advance our understanding of the relationship between DT traits and depressive symptoms in order to address previous limitations found in the literature. A better knowledge of the mechanisms and factors explaining this relationship would help to detect more rapidly the emotional problems associated with DT traits and could inform the development of improved intervention programs (*Harrop et al., 2017*; *Nock et al., 2010*).

First, the gender differences observed for the total scores and sub-dimensions of DT traits provide partial support for Hypothesis 1. The results for psychopathy and Machiavellianism traits are in line with those found in the previous literature (*Muris et al., 2017*; *Jonason & Davis, 2018*; *Cale & Lilienfeld, 2002*; *Krampen et al., 1990*). The higher psychopathy scores found in men in comparison with women are compatible with scientific evidence showing that in general men present more antisocial behaviors due to social or genetic factors (*Cale & Lilienfeld, 2002*; *Book, Starzyk & Quinsey, 2001*; *Reidy et al., 2009*). The higher scores that men showed in Machiavellianism could be explained by psychological sex-role variables such as normative sex-role orientations (normative behaviors for women and for men), and gender-related self-concepts (self-perceptions of masculinity and femininity), since both the role and behavior associated with the masculine gender are more strongly linked with Machiavellian traits (*Krampen et al., 1990*). On the other hand, contrary to the findings revealed by the meta-analysis conducted by *Grijalva et al. (2015)*, gender differences were not found in total narcissism. This lack of significant differences could be due to the general increase of narcissism in current

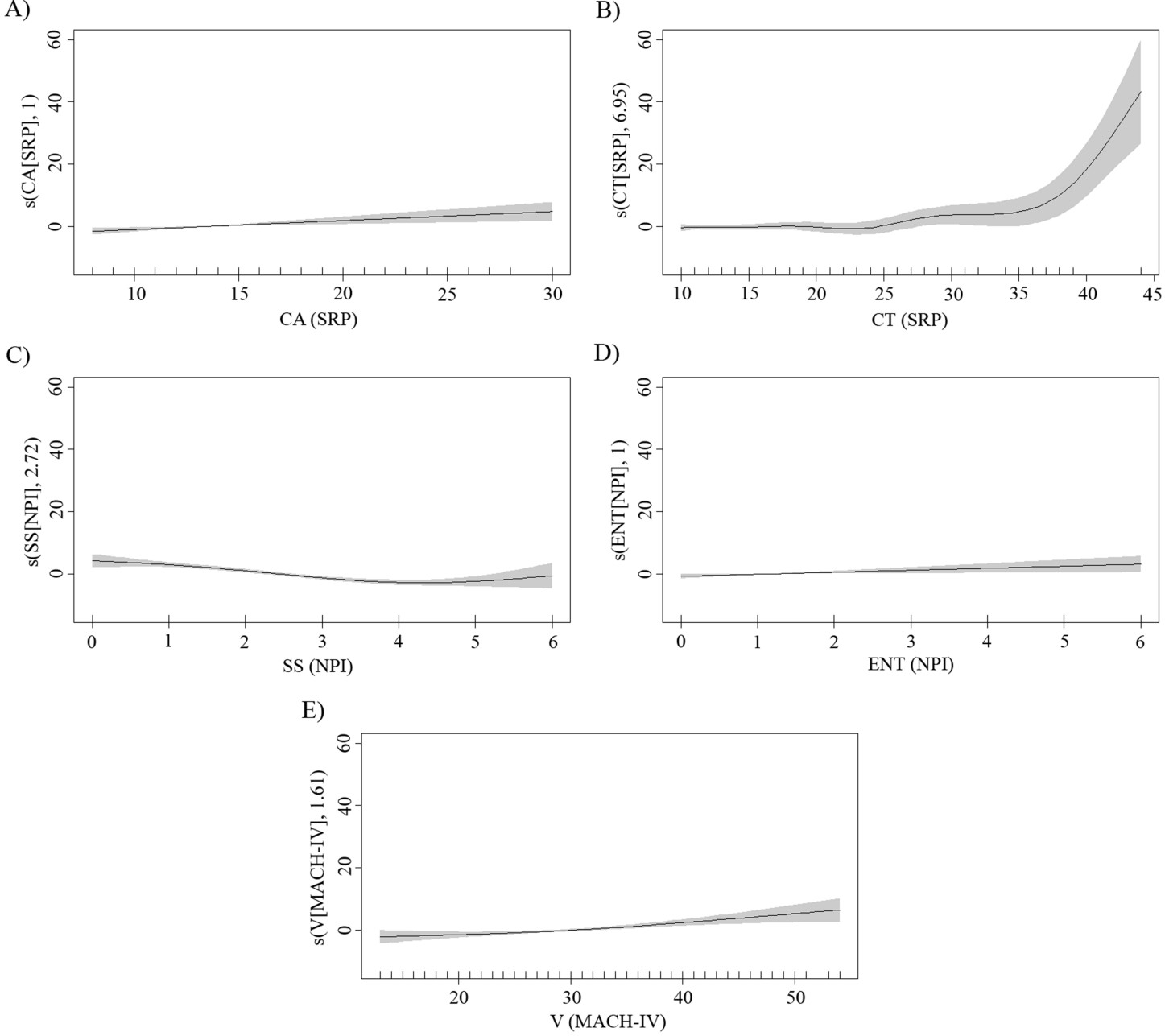

**Figure 1 Plots showing the smooth components of the fitted GAM.** The solid black line denotes the relationship between significant DT sub-dimensions and depressive symptoms estimated from the GAM. The gray shaded area indicates 95% confidence intervals. The estimated degrees of freedom of the smooth curves are shown in parentheses on $y$-axis. (A) Callous affect (CA); (B) criminal tendencies (CT); (C) self-sufficiency (SS); (D) entitlement (ENT); (E) cynical view of human nature (V).

society (both for men and women; *Twenge & Campbell, 2008*). Our study only showed differences in the vanity and entitlement sub-dimension of narcissism, which is in agreement with the previous literature (*Grijalva et al., 2015*).

In accord with Hypothesis 2, higher psychopathy and Machiavellianism scores (total score and all sub-dimensions) were related to higher levels of depressive symptoms.

Regarding psychopathy, there is currently an open debate concerning the relationship between this trait and depression, since some authors consider both constructs to be mutually exclusive (*Lovelace & Gannon, 1999*). This is due to the fact that psychopathy is understood as an activation disorder and, in contrast, mood dysfunctions such as depression are included among the inhibition disorders. Our findings are compatible with the results of previous studies supporting the notion that both constructs are not exclusive and that, in fact, there is a positive relationship between them (*Stinson, Becker & Tromp, 2005*; *Tokarev et al., 2017*). Several theories have emerged as a consequence of the need to explain these results. For example, research with adolescents has revealed that the externalization of negative moods such as anxiety or depression can result in antisocial and psychopathic behaviors (*Harrington, 2001*; *Kasen et al., 2001*; *Sayar, Ebrinc & Ak, 2001*). Another hypothesis refers to the different symptoms that define the concept of depression. Even though there are people with high psychopathic traits that do not show the emotional symptoms of depression, many of them still obtain high overall scores on this disorder since they present the physiological symptoms (*Stinson, Becker & Tromp, 2005*). With respect to Machiavellianism, the positive relationship found with depression is well supported by the previous literature (*Al Aïn et al., 2013*). The Machiavellian concept is largely characterized by the emotional detachment that is usually observed in people who suffer from depression (*Demenescu et al., 2010*). Moreover, Machiavellianism is related to anhedonia or the inability to feel pleasure, a symptom that is frequently found in depressive disorders (*Treadway & Zald, 2011*).

In relation to narcissism, the lack of a relationship between the total score of this trait and depression is in accord with Hypothesis 3 and previous findings in the literature (*Jonason et al., 2015*). Narcissism is a trait that differs from psychopathy and Machiavellianism in terms of its social character, where the individual tends to seek the validation of others (*Raskin & Terry, 1988*). Thus, individuals with high scores on this trait often have good social support (*Jonason & Schmitt, 2012*), which is considered to be a protective factor for mood disorders (*Almagiá, 2014*). However, although no significant results were found with total score, we found that depressive symptoms showed a negative relationship with the self-sufficiency sub-dimension, and a positive relationship with the entitlement sub-dimension. These results are unsurprising, given that self-sufficiency has previously been linked to healthier functioning, whilst entitlement has been linked to more maladaptive behaviors (*Watson et al., 1984*).

In relation to Hypothesis 4, some of the sub-dimensions of the DT traits associated with depressive symptoms showed significant differences depending on gender. The negative relationship between depressive symptoms and self-sufficiency (NPI) was stronger in women than men, the relationship with vanity (NPI) was only observed in women, whilst the relationship with disregard for conventional morality (MACH-IV) was only observed in men. There are many studies that have analyzed gender differences in both DT (*Paulhus & Williams, 2002*; *Muris et al., 2017*) and depression (*Girgus & Yang, 2015*; *Piccinelli & Wilkinson, 2000*; *Salk et al., 2016*) separately; however, to the best of our knowledge, our study is the first to explore the moderating effect of gender on the relationship between

these variables. The gender differences found in the relationship between vanity and depression could be explained by the fact that aspects of physical appearance that are included in the vanity sub-dimension are more dominant in women. In this gender, higher vanity scores are associated with greater general satisfaction with personal appearance (*Grijalva et al., 2015*). Thus, the result found in our study could be due to the negative relationship between body satisfaction and depressive symptoms (*Rentz-Fernandes et al., 2017*). Moreover, self-sufficiency in women is associated with greater economic independence and career success (*Wettersten et al., 2004*), which would have a strong negative association with depressive symptoms and greater life satisfaction (*Montesó-Curto, 2014*). Finally, in men, higher scores on disregard for conventional morality could be related to a lack of social connections, which could result in higher levels of depression, and in general, worse mental health (*Jonason et al., 2015*).

Particularly relevant to our aims were the findings observed in relation to Hypothesis 5. GAM analysis controlling for shared variability among the sub-dimensions of the DT traits (and gender effect) revealed that the sub-dimensions most strongly associated with depressive symptoms were callous affect (SRP-III), criminal tendencies (SRP-III), entitlement (NPI), self-sufficiency (NPI), and cynical view of human nature (MACH-IV). Thus, although previous correlation analyses showed an association between depression and a greater number of sub-dimensions, the effect of many of these was shared and can be explained by the sub-dimensions included in the GAM model. Working individually on each trait or sub-dimension could be effective, but, given the common core of characteristics shared, it makes more sense to pay particular attention to those sub-dimensions that were included in this model.

From an applied point of view, the findings of this study could have implications for the implementation of prevention and treatment programs focused on depression in individuals with high levels of DT traits. Research studying DT has usually focused on the psychological factors underlying this construct and the negative implications of these outcomes for society. However, rarely have researchers investigated the consequences that these traits may have for the individuals themselves, such as depressive symptoms. Our results suggest that the detection of high scores on certain sub-dimensions of the DT traits could be useful for informing the development of preventive interventions that would help to avoid future depressive symptoms.

Finally, it is important to address some of the limitations of this research. Although a high number of men participated in this study ($n = 197$), the sample was composed mainly of women. Future investigations should aim to use a more balanced sample. Moreover, it would be interesting to work with a clinical sample in order to replicate our results in individuals with a diagnosis of major depression based on psychiatric and clinical interviews (*Sheehan et al., 1998*). Finally, it must be kept in mind that the correlational analysis included in this study does not allow us to establish causal relationships. We have only focused on studying the association between both constructs. Additional prospective experimental studies would be needed to confirm the causal direction of the relationships found here.

## CONCLUSION

The present study explored the relationship between DT traits and depressive symptoms. Our findings revealed that there are a series of factors characterizing individuals with higher DT scores, which can be related to depression problems. The sub-dimensions of callous affect and criminal tendencies for psychopathy, cynical view of human nature for Machiavellianism, and entitlement and self-sufficiency for narcissism were the factors most strongly related to depressive symptoms. Furthermore, we observed how the effect of some of the DT sub-dimensions depended on gender. The results of this study could be exploited as a tool for designing prevention and intervention programs aimed at decreasing the negative consequences suffered by individuals with high DT scores. The reduction of depressive symptoms in this population can have important benefits for the individuals themselves, allowing for better integration in society and thus limiting the negative consequences of these traits for the community.

### Funding

This work was supported by the Spanish Ministry of Economy, Industry and Competitiveness (PSI2017-84170-R to Pablo Fernández Berrocal). The funders had no role in study design, data collection and analysis, decision to publish, or preparation of the manuscript.

### Grant Disclosures

The following grant information was disclosed by the authors:
Spanish Ministry of Economy, Industry and Competitiveness: PSI2017-84170-R to Pablo Fernández Berrocal.

### Competing Interests

The authors declare that they have no competing interests.

### Author Contributions

- Raquel Gómez-Leal conceived and designed the experiments, performed the experiments, analyzed the data, contributed reagents/materials/analysis tools, prepared figures and/or tables, authored or reviewed drafts of the paper, approved the final draft.
- Alberto Megías-Robles conceived and designed the experiments, performed the experiments, analyzed the data, contributed reagents/materials/analysis tools, prepared figures and/or tables, authored or reviewed drafts of the paper, approved the final draft.
- María José Gutiérrez-Cobo conceived and designed the experiments, authored or reviewed drafts of the paper, approved the final draft.
- Rosario Cabello conceived and designed the experiments, authored or reviewed drafts of the paper, approved the final draft.
- Enrique G. Fernández-Abascal conceived and designed the experiments, performed the experiments, authored or reviewed drafts of the paper, approved the final draft.

**PeerJ** ______________________________________

- Pablo Fernández-Berrocal conceived and designed the experiments, performed the experiments, analyzed the data, contributed reagents/materials/analysis tools, prepared figures and/or tables, authored or reviewed drafts of the paper, approved the final draft.

## Human Ethics

The following information was supplied relating to ethical approvals (i.e., approving body and any reference numbers):

The Research Ethics Committee of the University of Málaga approved the study protocol as part of the project SEJ-07325 (approval number 10-2018-H).

## Data Availability

The raw data is available in the Supplemental File.

## Supplemental Information

Supplemental information for this article can be found online at http://dx.doi.org/10.7717/peerj.8120#supplemental-information.

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
