# Peer review of "Relationship between the Dark Triad and depressive symptoms"

_PeerJ, doi:10.7717/peerj.8120_

## Round 0.1 · original submission · Major Revisions

This is an interesting study on the DT-depression association. In my opinion, the study has the below issues and therefore a substantial revision is required.

1. Introduction. It remains unclear why DT results in depression, this is very important because the main hypothesis is a path from DT to depression. For me, depressive symptoms can be viewed as a response of negative life events and the direct result of these events. I think it is difficult to speculate that DT directly causes depression, so is there a direct path between DT and depression? If not, the study can not analyse the data in this manner. The authors also mixed depressive symptoms and depressive disorders. The study focused on depressive symptoms, in fact, not disorders.
2. I do not agree with the authors that the data of this study can be used to establish predictive model. To predict, the study design must be prospective, but this is a cross-sectional study, which can only tests correlations.
3. As the authors mentioned, depressive symptoms are the result of multiple factors, including personality. For me, personality is stable trait, but symptoms are time- and situation-dependent. So is it reasonable to hypothesize that DT personality results in depression and gender moderates this path? Further, the so-called predictive model of depression only included DT domains, can this model be used to predict depression?
4. The study sample's representativeness is not good, particularly a small proportion of men.
5. BDI-II is also a multi-dimension scale. Why not examining the relationship between the constructs of depression and DT? Further, examining the association between manifest variables of depression and DT is not a good statistical approach, because these variables have their constructs and these constructs are correlated. Latent variables of these constructs and SEM would be preferred. Results from such analyses should be more informative.

Please address all the comments of the reviewers in an appropriate rebuttal.

Reviewer 1 ·

Basic reporting

The Manuscript entitled “Relationship between the dark triad and depression: The moderating effect of gender” provides interesting findings on the relationships between the subclinical personality traits of the dark triad and Depression symptoms in a quite large sample. However, there are several things which definitely need to be changed before I can approve publication. The most important point is about the statistical analyses and their interpretation, specifically, including gender into the regression models.
Please see my following review below.

Overall, the article is clearly written, mostly easy to follow and interesting to read.
1. However, the DT traits are sometimes written starting with a capital letter (e.g. in the abstract) and sometimes starting with a lowercase letter. I would suggest to spell the terms as follows throughout the manuscript: “Machiavellianism”, “psychopathy” and “narcissism”. This is in line with the classic terminology / spelling in scientific papers.
2. Moreover, while I do understand that the first paragraph of the introduction should be seen as a general introduction into the topic, I am still of the opinion that the associations with other traits, emotional deficits, etc. is depicted (too) broadly. It is well known that there are important differences between the three DT traits in the associations with other constructs. Hence, it would be advisable to not generalize such associations across all three DT traits (as it is done so far in paragraph 1).
3. Additionally, I am of the opinion that the introduction would be easier to read if the DT traits would be explained before referring to their deficits (hence, one might want to think about changing the order of paragraphs 3 and 4 in the introduction).
4. The sentence in lines 150-151 is not in line with the actual sample used in the present work. Actually, the present sample shows rather low scores in the DT traits. Therefore, it should not be mentioned that people scoring high in DT should be analyzed. The same is true for hypothesis 5.
5. Lastly, I do not understand, why hypothesis 4 is only related to sub-dimensions of the DT traits.

Experimental design

The present study is a self-report association study, the research question(s) are mostly well defined and also the gap in the previous literature justifying this research is well explained. However, I have several important comments on the methods.
1. It would be advisable to include information on the consent (written / informed / …) in the manuscript.
2. Lines 165-168: The information on the sample are not in line with the information from the SPSS file (e.g. in the SPSS file there are 158 instead of 148 males). Please check all numbers throughout the manuscript carefully.
3. From what is written in the method section right now, it does not get clear whether the Cronbach’s alphas are derived from the cited studies or from the sample of the present study. It would be great if the alphas derived from the present sample would be presented.
4. The statement “Third, we explored the possible moderating effect of gender on the significant relationships found in the previous step.” (lines 220-221) is actually wrong. With NONE of the analyses implemented, actual moderation effects of gender were investigated. Actually, not even the correlations between males and females were statistically tested for significant differences (e.g. by a Fisher’s z-Test). To explore moderation effects, it would be best to include gender as well as the gender x DT (sub-)dimension interaction terms alongside the other predictors (main effects of DT (sub-)dimensions) into one regression model. If this is not done, no moderation effects are investigated. Therefore, also the interpretations and even the title are not correct.
5. However, when looking at the data, it becomes obvious that several of the scales under investigation are not normally distributed. This is especially also true for the BDI-II. Hence, parametric testing does not seem to be appropriate for these data. I strongly suggest to use statistic tests (throughout the complete manuscript) which to not rely on the assumption of normality.
6. Moreover, reporting effect sizes is very good. However, given the different sample sizes of males and females, the authors might want to think about not reporting Cohen’s d but a corrected effect size (e.g. Hedge’s g) in Table 1.

Validity of the findings

The results are interesting. However, next to the comments on the statistical analyses, I also have some comments on the presentation of the results as well as – especially – regarding their interpretation (these comments built upon the comments on the statistical analyses mentioned above).

1. In Table 1 it appears as if the scores of the total scales of the questionnaires would be calculated by building sum scores; however, subscales were built by calculating mean scores. For reasons of consistency I would strongly suggest to use either sum or mean scores for all (sub-)dimensions.
2. In Table 1: Females score (descriptively) higher in VAN (NPI) compared to males. Therefore, the t-value should be positive. In line with this, please check the numbers in Table 1 (and also the other tables) carefully for errors (e.g. rounding errors) and correct them when necessary. When working with the SPSS-File provided by the authors, I sometimes receive (slightly) different values.
3. Table 1: In the note below Table 1, “SD” is explained by “standard error”. However, SD stands for “standard deviation”. Please carefully check which of the two is reported in the table and change the respective parts / descriptions, accordingly.
4. I would suggest to use the same order of the subscales when presenting methods / results throughout the manuscript (e.g. compare description of questionnaires vs. Table 1 vs. Table 2).
5. In the text (in the results section) about the specific results, I would suggest to write the complete names of the scales; otherwise it is hard to follow / understand results.
6. As mentioned previously, to examine (and interpret) moderation effects of gender, this factor MUST be included in the regression models (alongside main effects of the DT (sub-)dimensions and the respective interaction effects).
7. Lines 269-270: Actually, Hypothesis 1 is not fully supported. Because in the introduction this hypothesis is referring to all DT traits; however, gender differences were not found in each if the (sub-)dimensions / traits.
8. Lines 269-274: There is much more and newer research about gender differences in these traits. Hence, also newer literature should be cited.
9. Lines 686-688: Please include references.
10. Lines 290-292: Please cite more than one reference if you are referring to “studies”.
11. 322-323: Please cite more than one reference if you are referring to “many studies".
12. If gender (and the respective interaction terms) are not included in the regression model, actually no moderating effects were tested; Just because some correlations reached significance in one (e.g. the female) sample but not the other (e.g. the male) sample, it does not necessarily has to be due to a moderation effect of gender; instead the differences in significance(s) could also be due to the different sample sizes (in line with this, the differences between the correlations do not seem to be very big). Also, descriptively the sizes of the correlations do not seem to be very big. Hence, the complete paragraph on hypothesis 4 in the discussion (and the conclusion) is not really correct (see already comment above). Only if gender alongside the other predictors (as well as the interaction terms etc.) is included in the one regression model, potential moderating effects can be discussed.

Additional comments

I am sure the authors can tackle most of the above mentioned comments. In conclusion and most important, the authors should think about the specific statistical analyses and, hence, also the discussion of their results. Also potential errors in the reported results should be corrected.
Lastly, I have to minor comments:
1. Why do the author’s mention their work would be an “original Review” in their comment? I guess it should mean “original research”.
2. Line 253: It is written “VA” but the scale is actually labeled: VAN. Please correct.

·

Basic reporting

The authors investigated the associations between DT traits, depression and gender among a larger sample of participants in early adulthood. The authors observed that specific dimensions of DT traits were associated with symptoms of depression and gender. The authors further underlined that DT traits appeared to be associated with difficulties in emotion regulation.

Abstract: Please add gender ratio.
“…..DT sub-dimensions depending on gender..”; ok, but on which gender specifically?
Introduction: Generally very well written; references are timely, though, the authors might balance the disadvantages of DT traits against the advantages of DT traits; this holds particularly true, as from the view point of evolutionary psychology and psychiatry, DT would have been sorted out over time, which, however, was not the case. Further, give a closer look at (Sabouri, Gerber, Lemola, et al., 2016; Sabouri, Gerber, Sadeghi Bahmani, et al., 2016).
“Depression is one of the most common mental disorders worldwide.”; please add references and, most importantly, report gender differences.
When describing the DSM 5 criteria for major depressive disorders, please add the time frame and clearly distinguish between the diagnosis of major depressive disorders and symptoms of depression. This is particularly important, as from the Method section it turned out that ‘depression’ was solely self-reported and not based on a thorough psychiatric and clinical interview, for instance with the MINI (Sheehan et al., 1998).
“…but relatively little importance has been assigned to the negative emotional consequences that these people suffer.”; see (Sabouri, Gerber, Lemola, et al., 2016)
“….variables such as SYMPTOMS OF depression (Jones & Paulhus, 2017).”
Collectively, the Introduction was particularly well written; the reader gets a comprehensive overview of the state-of-the-art; hypotheses were theory-based and well formulated. Congrats on you.
Methods:
Report, how you did make sure that participants answered correctly to the items. How did you handle missing data? Report, how much time participants needed on average to complete the questionnaires.
SPSS® version 20.0 (IBM Corporation, Armonk NY, USA).
Report the alpha level of significance.
Results:
“…for the ENT…”; I suggest to spelling out all abbreviations; don’t count too much on the reader’s working memory.
Multiple regression analysis (and Table 3); report in more details, if basic statistical requirements to run a multiple regression analysis were met (Durbin-Watson coefficients; multicollinearity; homoscedasticity); Table 3; it appears that all subdimensions statistically significantly contributed to explain the outcome variable; please check once again very carefully, as such results are rather surprising; further, a t-value of 1.86 most probably was not statistically significant.
Discussion: The authors discussed very well their data.
Conclusion: Although it is a matter of taste, I suggest to substantively shorten the Conclusion section.


References
Sabouri, S., Gerber, M., Lemola, S., Becker, S. P., Shamsi, M., Shakouri, Z., . . . Brand, S. (2016). Examining Dark Triad traits in relation to sleep disturbances, anxiety sensitivity and intolerance of uncertainty in young adults. Compr Psychiatry, 68, 103-110. doi:10.1016/j.comppsych.2016.03.012
Sabouri, S., Gerber, M., Sadeghi Bahmani, D., Lemola, S., Clough, P. J., Kalak, N., . . . Brand, S. (2016). Examining Dark Triad traits in relation to mental toughness and physical activity in young adults. Neuropsychiatr Dis Treat, 12, 229-235.
Sheehan, D. V., Lecrubier, Y., Sheehan, K. H., Amorim, P., Janavs, J., Weiller, E., . . . Dunbar, G. C. (1998). The Mini-International Neuropsychiatric Interview (M.I.N.I.): the development and validation of a structured diagnostic psychiatric interview for DSM-IV and ICD-10. J Clin Psychiatry, 59 Suppl 20, 22-33;quiz 34-57.

Experimental design

See above

Validity of the findings

See above

Additional comments

See above

---

## Round 0.2 · Minor Revisions

Thank you for the revisions. Some revisions are needed for the further process of this paper. Please revise it accordingly.

Reviewer 1 ·

Basic reporting

The manuscript titled “Relationship between the dark triad and depressive symptoms” by Gómez-Leal et al. improved a lot and the authors did a great job in the revising the manuscript. It reads very well, the introduction, rationale and hypotheses are easy to follow and understand, and the statistical methods are appropriate. The discussion is very interesting.
I only have some minor comments / suggestions on the manuscript (word-file). Afterwards, I approve publication.


Basic Reporting:

Abstract: “The aim of the present study was to analyse the relationship between the DT and depressive symptoms in order to identify those factors most strongly associated with the development of depression in individuals scoring high on DT.“: I think it would be advisable to include some information on the sub-dimensions in this sentence. Because the sub-dimensions have – as far as I understand – not been investigated in this context so far: “The aim of the present study was to analyse the relationship between the DT including sub-dimensions and depressive symptoms in order to identify those factors most strongly associated…”.

Lines 108-109: “However, relatively little importance has been assigned to the negative emotional consequences that these individuals may suffer.”: I guess by writing “these individuals” it is referred to individuals scoring high in the DT? Please clarify.

Lines 201-202: “A score of 0-13 indicates minimal depression, ...”: It would be great if a reference for these cut-off scores would be cited. In the past, I realized that several different rules for the cut-off scores of the BDI-II exist.

Experimental design

Experimental design:

I acknowledge that Cronbach’s alphas are reported for the present sample. However, it seems like these values are only reported for the complete scales of the questionnaires. As the subscales are also of importance for the present study, it would be advisable to also report Cronbach’s alphas for the subscales. At least it might be possible to give some information on the range in which the Cronbach’s alphas of the subscales lie.

Line 243-244: “Descriptive statistics, t-tests, Spearman’s correlations and Fisher's Z-tests were all conducted using the SPSS® version 23.0”: I guess instead of “t-test” it should be Mann-Whitney U test. Is this correct? Additionally, I am pretty sure that it is not possible to perform Fisher’s z-Tests in order to compare two correlations in SPSS. Please clarify.

Lines 255-256: “We included in the model those DT sub-dimensions that were significant in the correlational analysis”: Was the significance checked in the complete sample or the male / female sample? Please clarify.

Additionally, it would be very helpful if some more information on the GAM would be given. There are several settings when implementing a GAM, which can / must be changed according to the data. Also, as the scores assessing the predictors (DT traits and sub-dimensions) have different ranges (depending on the items included to calculate the score, the response scale, …), I am wondering whether it is necessary to standardize these values. Maybe some more information on this topic can be given. At least it would be helpfull to discuss this issue in the discussion section of the manuscript.

Validity of the findings

Validity of the findings:

Caption Table 2: „ Spearman's correlations between sub-dimensions of DT and depression.“: Correlations with the total scores are also presented in Table 2. Therefore, it might be a good idea to think about deleting the „sub“ before the „-dimensions“ or to put it into parentheses.

Lines 284-288: “When specific gender comparisons were made with respect to the correlations using the Fisher's z-test, it was found that men, compared with women, showed a stronger negative relationship between the narcissism sub-dimensions of self-sufficiency (Z = 2.05, p < .05) and vanity (Z = 2.31, p < .05) with depressive symptoms.”: When looking at Table 2, it seems like actually women show stronger negative correlations than men. Please clarify.

Line 293: “…revealed six significant sub-dimensions…”: As far as I can see, it should be five sub-dimensions.

In the data analyses section, it is stated that gender was also included in the GAM. It would be great to clarify i) whether only the main effect of gender was included or also interaction effects with gender, and ii) whether a gender effect turned out to be significant or not.

Additional comments

Line 400: “… included in the regression model.”: I guess it should be “GAM” (instead of regression model). Please clarify.

Please check all statics – and especially information on significance – carefully, once again. For example, when I calculate the mean age of the sample, I receive M=35.757, which would be rounded to 35.76 years (instead of 35.75). Moreover, when running a Mann-Whitney U test, I receive a p-value of .056 for the gender difference in the CA scale of the SRP-III. However, in Table 1 there is an asterisk behind the number, which would indicate significance. The p-value for the gender difference in the T scale of the Mach-IV is actually 0.009767, hence, two asterisks rather than one would be appropriate. The gender differences in the ENT and VAN scales of then NPI show p-values of .008 and .005, which would need to be indicated with two asterisks rather than 1. Of note, these are just some examples.

·

Basic reporting

In my opinion, the authors took their job to revise the ms very seriously. Congrats on you! The authors have dealt with every issue raised by the editorial board and the reviewers in a thorough and comprehensive fashion. Overall, the ms is in a very nice shape, the content is interesting, references are timely, and the text has a good flow.

Experimental design

see above; no further comments

Validity of the findings

see above; no further comments

Additional comments

see above; no further comments

---

## Round 0.3 · accepted · Accept

Thanks for your revisions.

Reviewer 1 ·

Basic reporting

The manuscript "Relationship between the dark triad and depressive symptoms" by Gómez-Leal et al. improved a lot and the authors did a great job in the revising the manuscript, again.
Overall, the manuscript reads very well, the introduction, rationale and hypotheses are easy to follow and understand, and the statistical methods are appropriate. The discussion is very interesting.
I do not have further comments and agree on publication of the manuscript.

Experimental design

-

Validity of the findings

-

Additional comments

-